# Establishment of the Foot-and-Mouth Disease Virus Type Asia1 Expressing the HiBiT Protein: A Useful Tool for a NanoBiT Split Luciferase Assay

**DOI:** 10.3390/v16071002

**Published:** 2024-06-21

**Authors:** Giyoun Cho, Hyejin Kim, Dong-Wan Kim, Seong Yun Hwang, Ji-Hyeon Hwang, Yeon Rae Chae, Yoon-Hee Lee, Ok-Mi Jeong, Jung-Won Park, Sung-Han Park, Jong-Hyeon Park

**Affiliations:** Center for Foot-and-Mouth Disease Vaccine Research, Animal and Plant Quarantine Agency, 177, Hyeoksin 8-ro, Gimcheon-si 39660, Gyeongsangbuk-do, Republic of Korea; libretto@korea.kr (G.C.); hyejin86@korea.kr (H.K.); dongwan12@korea.kr (D.-W.K.); hsy8592@korea.kr (S.Y.H.); jihyeonh87@korea.kr (J.-H.H.); codusfo1@korea.kr (Y.R.C.); lyhee74@korea.kr (Y.-H.L.); nabee@korea.kr (O.-M.J.); parkjw6254@korea.kr (J.-W.P.)

**Keywords:** FMDV, NanoBiT, HiBiT, LgBiT, Luciferase assay

## Abstract

Foot-and-mouth disease virus (FMDV) is a highly contagious virus that affects cloven-hoofed animals and causes severe economic losses in the livestock industry. Given that this high-risk pathogen has to be handled in a biosafety level (BSL)-3 facility for safety reasons and the limited availability of BSL-3 laboratories, experiments on FMDV call for more attention. Therefore, we aimed to develop an FMDV experimental model that can be handled in BSL-2 laboratories. The NanoBiT luciferase (Nano-luc) assay is a well-known assay for studying protein–protein interactions. To apply the NanoBiT split luciferase assay to the diagnosis and evaluation of FMD, we developed an inactivated HiBiT-tagged Asia1 Shamir FMDV (AS-HiBiT), a recombinant Asia1 shamir FMDV with HiBiT attached to the VP1 region of Asia1 shamir FMDV. In addition, we established LgBiT-expressing LF-BK cell lines, termed LgBit-LF-BK cells. It was confirmed that inactivated AS-HiBiT infected LgBiT-LF-BK cells and produced a luminescence signal by binding to the intracellular LgBiT of LgBiT-LF-BK cells. In addition, the luminescence signal became stronger as the number of LgBiT-LF-BK cells increased or the concentration of inactivated AS-HiBiT increased. Moreover, we confirmed that inactivated AS-HiBiT can detect seroconversion in sera positive for FMDV-neutralizing antibodies. This NanoBiT split luciferase assay system can be used for the diagnosis and evaluation of FMD and expanded to FMD-like virus models to facilitate the evaluation of FMDV vaccines and antibodies.

## 1. Introduction

Foot-and-mouth disease (FMD) is one of the most infectious diseases that causes severe economic losses in the livestock industry [1]. The causative pathogen, FMD virus (FMDV), which belongs to the genus *Apthovirus* of the family *Picornaviridae*, infects cloven-hoofed animals, including various ruminants and wildlife animals [2]. There are seven FMDV serotypes: O, A, C, SAT1, SAT2, SAT3, and Asia1 [3,4]. FMDV has a positive-sense, single-stranded RNA genome that encodes four structural proteins (VP4, VP2, VP3, and VP1), which form the infectious spherical particles, and eight non-structural proteins (L, 2A, 2B, 2C, 3A, 3B, 3C, and 3D) [5,6]. VP1, VP2, and VP3 are surface-exposed proteins, whereas VP4 is internally located [7,8]. The GH loop, formed by residues 140–160 of VP1, is the major neutralizing antigenic site and induces humoral immunity [9,10,11]. Because of the serious consequences of its laboratory spread and its high risk to animal health and national agricultural economies, FMD is designated as a Containment Group 4 animal pathogen in the 2023 WOAH Manual of Diagnostic Tests and Vaccines for Terrestrial Animals [12]. In Korea, FMD is a “Contagious animal disease Type I” according to the “Act On The Prevention Of Contagious Animal Diseases” [13]. The requirement for a biosafety level (BSL)-3 facility for handling FMDV hinders extensive research of the virus [14]. BSL-3 laboratories are not costly to build, but are complex to manage. Therefore, experimental methods for the serological diagnosis of FMDV that avoid the need for BSL-3 facilities are needed. The inactivation of FMDV allows experimental handling of the virus in a BSL-2 laboratory. Inactivated FMDV can be safely handled and avoids biosafety restrictions.

Biosensors have proven useful in the generation of serological assays. Nano-luciferase (Nano-Luc)-based biosensors have been applied to several viruses, including the SARS-CoV-2 virus, dengue virus, Japanese encephalitis virus, hepatitis C virus, and FMDV as well [15,16]. Nano-Luc has been cleaved between residues 156 and 157, generating a large (18 kDa) subunit termed “LgBiT” and a small (1.3 kDa) subunit of 11 amino acids termed “HiBiT” [17,18]. HiBiT has a high affinity for LgBiT, which makes it ideal for protein tagging, and because of the small size of HiBiT, HiBiT-tagged proteins largely maintain normal function [19]. Because the HiBiT-tagged proteins can be easily detected and quantified based on luminescence using the Nano-Glo luciferase assay system, the HiBiT-LgBiT biosensor technology offers high sensitivity, the convenience of a single reagent addition step, and a short time [20].

In this study, we established a system to identify FMDV entry in cells using a Nano-Luc biosensor. In particular, the HiBiT subunit was inserted into the VP1 GH loop of FMDV Asia1 Shamir to generate HiBiT-tagged FMDV (AS-HiBiT). LF-BK cell lines stably expressing the LgBiT protein were developed and named “LgBiT-LF-BK”. We showed that inactivated AS:HiBiT can attach to and enter LgBiT-LF-BK cells, producing luminescence signals, and that anti-Asia1 sera with FMDV-neutralizing antibodies (Abs) can inhibit AS-HiBiT entry into LgBiT-LF-BK cells. Furthermore, we demonstrated that our Nano-Luc system can be applied as a tool for evaluating viral propagation and identifying anti-FMDV sera in BSL-2 facilities.

## 2. Materials and Methods

### 2.1. Cells

Baby hamster kidney (BHK)/T7-9 cells stably expressing the T7 RNA polymerase were provided by Dr. Ito, Gifu University, Japan [21]. ZZ-R 127 fetal goat tongue cells and LF-BK fetal porcine kidney cells were provided by the Friedrich Loeffler Institute (Greifswald, Germany) and Plum Island Animal Disease Center (Greenport, NY, USA), respectively [22,23]. BHK-T7-9 cells were maintained in Glasgow Minimum Essential Medium (Gibco BRL, Paisley, UK) supplemented with 10% fetal bovine serum (FBS; Gibco) and 10% tryptose phosphate broth (Sigma-Aldrich, St. Louis, MO, USA). ZZ-R cells were maintained in Dulbecco’s Modified Eagle’s Medium/F12 (DMEM-F12; Corning, Union City, NJ, USA) supplemented with 10% FBS and 1% penicillin–streptomycin solution (P/S: Corning, Manassas, VA, USA). LF-BK cells were maintained in DMEM supplemented with 10% FBS (Gibco) and 1% P/S (Corning). The cells were maintained in a 5% CO_2_ atmosphere at 37 °C.

### 2.2. Production of Anti-Sera from Guinea Pigs

The animal experiments were approved by the Institution Animal Care and Use Committee of the Animal and Plant Quarantine Agency (APQA) of the Republic of Korea (approval No. 2024-860). Serum samples with high levels of neutralizing Abs against FMDV Asia1 were prepared by inoculating six-week-old female guinea pigs (*n* = 3) with a commercial FMD vaccine (0.2 mL/dose) containing O Manisa, O 3039, A22 Iraq, and Asia1 Shamir (Boehringer Ingelheim, RP, DEU). Anti-sera samples were collected at 0 (negative serum) and 28 days (positive serum) post-vaccination (dpv). The anti-sera were inactivated in a heat block at 56 °C for 30 min and then analyzed using an FMDV Asia1 Ab enzyme-linked immunosorbent assay (ELISA) and virus neutralization test (VNT). A PrioCHECK FMDV Asia1 Ab ELISA Kit (Prionics AG, Zurich, Switzerland) was used to detect structural protein (SP) Abs in the sera. The absorbance of the ELISA plate was converted to a percentage inhibition (PI) value. An animal was considered positive when the PI value was ≥50%. The VNT was performed according to the WOAH manual for terrestrial animals [12]. Virus neutralization (VN) titers were calculated as the reciprocal of the serum dilution neutralizing 100 TCID50 (50% tissue culture infective doses of the virus in 50% of the wells). The cut-off used in the VNT (1.65 log10) was based on the standard operating procedures of the World Reference Laboratory for FMD (Pirbright Institute, UK) [24].

### 2.3. Production and Characterization of HiBiT-Tagged FMDV Asia1 Shamir (AS-HiBiT)

The GSSG linker and HiBiT (ggcagcagcggcVSGWRLFKKISggcagcagcggc) coding sequences were inserted into the VP1 region of Asia1 Shamir (GenBank accession No. JF739177) and the modified complete genome of the Asia1 shamir was cloned into the pBluescript SK II vector to generate recombinant FMDV infectious clones [25]. Full-length recombinant Asia1 Shamir plasmids were linearized using the restriction enzyme SpeI (NEB, Ipswich, MA, USA), and the linearized plasmids were transfected into BHK-T7-9 cells using Lipofectamine 3000 (Invitrogen, Carlsbad, CA, USA). After incubation in a 5% CO2 atmosphere at 37 °C for 48–72 h, the viruses were harvested in three freeze–thaw cycles and amplified in fresh ZZ-R cells. Then, the viruses were adapted to LF-BK cells in consecutive order by passaging in cell culture. The viruses were harvested and centrifuged at 1760× *g*, 4 °C for 20 min. To inactivate the viruses, 3 mM binary ethylenimine (Sigma-Aldrich) was added, and the mixture was incubated in a shaking incubator at 26 °C for 24 h. Residual binary ethylenimine was neutralized with 2% sodium thiosulfate (Daejung Chemicals, Siheung-si, Republic of Korea). All FMDV-related experiments were performed in a BSL-3 facility at the APQA. The resulting AS-HiBiT FMDVs were analyzed for the SP of Asia1 using a VDRG FMDV 3Diff/PAN Ag Rapid kit (MEDIAN diagnostics, Chuncheon-si, Republic of Korea) and for HiBiT expression by Western blotting.

### 2.4. Establishment of Stable LgBiT-Overexpressing LF-BK (LgBiT-LF-BK) Cell Lines

An antibiotic-killing experiment was performed to determine the minimum concentration of hygromycin B required to kill all LF-BK cells for 3 days. LF-BK cells were transiently transfected with an LgBiT expression vector (Promega, Madison, WI, USA) using FuGENE HD transfection reagent (Promega). After 48 h, the cells were treated with 0.7 mg/mL hygromycin B (Glentham Life Science, Corsham, UK) to kill non-transfected cells and obtain LgBiT stably expressing LF-BK (LgBiT-LF-BK) cells. The resulting LgBiT-LF-BK cell clones were analyzed for LgBiT expression level by Western blotting using an LgBiT monoclonal (m)Ab (Promega).

### 2.5. Western Blotting

Inactivated viruses were mixed with 4X Laemmli sample buffer (Bio-Rad Laborato-ries, Hercules, CA, USA) containing a reducing agent (Invitrogen). The samples were heated at 95 °C for 5 min. The proteins were separated on 4–20% Bis-Tris gels (Bio-Rad) and transferred onto a polyvinylidene difluoride membrane (Bio-Rad) using a Trans-Blot Turbo transfer device (Bio-Rad). The membrane was blocked with 2% skim milk in Tris-buffered saline containing 1% Tween-20 (TBS-T) at room temperature under shaking for 1 h, washed three times with TBS-T for 10 min, and incubated with appropriate primary Abs at 4 °C under shaking overnight. The next day, the membrane was washed three times with TBS-T and incubated with the appropriate secondary Ab at room temperature under shaking for 1 h. Ab–antigen complexes were visualized using ECL Western blotting substrate (Amersham, Buckinghamshire, UK) and an Azure C600 device (Azure Biosystems, Dublin, CA, USA). The following commercial target protein-specific Abs were used: anti-HiBiT mAb (Promega, Madison, WI, USA), anti-LgBiT mAb (Promega), rabbit anti-Mouse IgG Ab (Millipore, Billerica, MA, USA), and anti-FMDV VP2 mAb (MEDIAN Diagnostics).

### 2.6. Concentration Determination and Verification of FMDV Antigen

Inactivated viruses were treated with 7.5% Polyethylene Glycol 6000 (Sigma-Aldrich) and 0.5 M NaCl (Sigma-Aldrich) at 4 °C for 16 h. The mixtures were centrifuged at 10,000× *g*, 4 °C for 30 min and the precipitates were resuspended in 1 mL of 20 Mm Tris buffer containing 300 mM KCl. The resuspended mixtures were centrifuged at 10,000× *g*, 4 °C for 10 min. The supernatant was layered on top of a 15–45% sucrose gradient and ultra-centrifuged at 10,000× *g* for 4 h. The 30–40% sucrose layers were collected and ultra-centrifuged at 10,000× *g* for 4 h. The pellet was resuspended using Tris buffer containing KCl buffer (pH 7.6) to eliminate residual sucrose at 4 °C. Viral particles were examined using transmission electron microscopy (TEM; H-7100FA; Hitachi, Tokyo, Japan).

### 2.7. Detection of AS-HiBiT Entry Using the NanoBiT Assay

LgBiT-LF-BK cells were seeded at 2 × 10^4^ cells/well into poly-D-lysin (Gibco)-coated white 96-well plates (Corning) and incubated at 37 °C for one day. After two washes with 1× PBS, inactivated AS-HiBiT viruses were added into each well and the plates were incubated at 37 °C for 1 h. The wells were washed with 1× PBS two times, and serum-free DMEM was added. Next, Nano-Glo luciferase (Luc) assay substrate (Promega) diluted in Nano-Glo Luc assay buffer was added to the wells and the luciferase signals were measured. To apply the NanoBiT assay to serological testing for the detection of FMDV SP Ab, guinea pig sera were mixed with one volume of inactivated AS-HiBiT and incubated at 37 °C for 1 h. Then, the serum/AS-HiBiT mixtures were added to the white 96-well plates seeded with LgBiT-LF-BK cells (2 × 10^4^ cells/well), and the plates were incubated at 37 °C for 1 h. The plates were washed twice with 1× PBS and serum-free DMEM was added. Luciferase activity was measured using the Nano-Glo Luc Assay System (Promega).

### 2.8. Statistical Analysis

All values are presented as mean ± standard error of the mean (SEM). Individual variances were computed for each comparison, and means were compared using one-way or two-way analysis of variance (ANOVA) followed by Tukey’s or Sidak’s multiple comparisons tests, using the GraphPad Prism (version 8; GraphPad Software, San Diego, CA, USA). *, *p* < 0.05; **, *p* < 0.01; ***, *p* < 0.001; ****, *p* < 0.0001; and ns, *p* > 0.05 (not significant). 

## 3. Results

### 3.1. Design of a Split Nano-Luc Complementation-Based FMDV Entry Technology

The two Nano-Luc components, HiBiT and LgBiT, can bind to generate luminescence signals. HiBiT-tagged FMDV can bind to the surface of LgBiT-expressing cells and subsequently be internalized, leading to NanoBiT split luciferase activity. In contrast, HiBiT-non-tagged FMDV cannot produce luminescence signals in LgBiT-expressing cells (Figure 1).

### 3.2. Production and Characterization of AS-HiBiT

To test our hypothesis, HiBiT was inserted into the GH loop (at amino acid 153) formed by residues 140–160 of the capsid protein VP1, located on the surface of the viral particle (Figure 2A). The plasmid was transfected into BHK-T7-9 cells to produce infectious viruses. The viruses were amplified in ZZ-R cells and subsequently adapted to LF-BK cells. AS-HiBiT was analyzed for protein expression using an FMDV 3Diff/PAN Rapid kit, Western blotting, and TEM (Figure 2B–D). The FMDV 3Diff/PAN Ag Rapid kit result is considered positive for FMDV Asia1 when a red line appears at the control line (C) and AS test line of the 3Diff strip, and at the C line and PAN test line of the PAN strip. Inactivated AS-HiBiT produced red lines at both lines of the 3Diff and PAN strips (Figure 2B). As expected, while inactivated AS-HiBiT produced positive signals when detected with HiBiT mAb and VP2 Ab, the wild-type virus produced no HiBiT protein band (Figure 2C). TEM revealed that inactivated AS or AS-HiBiT had a spherical shape with a 25–30 nm diameter (Figure 2D). To investigate whether inactivated AS-HiBiT could bind and enter LF-BK or LgBiT-transfected LF-BK cells, the LgBiT expression plasmid was transfected or inactivated AS-HiBiT was inoculated into LF-BK or LgBiT-transfected LF-BK cells (Figure 2E). Harvested whole-cell lysates were analyzed by Western blotting. Cells transfected with the LgBiT expression vector produced only the anti-LgBiT band. Cells inoculated with inactivated AS-HiBiT produced only the anti-HiBiT band. However, cells transfected with the LgBiT expression vector and inoculated with inactivated AS-HiBiT produced both the anti-LgBiT and anti-HiBiT bands, verifying the infectivity of inactivated AS-HiBiT (Figure 2E). To test the functionality of the HiBiT tag, LgBiT expression vectors were transfected into LF-BK cells, which were subsequently inoculated with inactivated AS-HiBiT. After incubation with luciferase substrate, luminescence signals were detected in the cells, indicating that LgBiT and the HiBiT tag of inactivated AS-HiBiT formed a functional NanoBiT split luciferase (Figure 2F).

### 3.3. Establishment and Characterization of LF-BK Cell Lines Stably Expressing LgBiT

To establish cells stably expressing LgBiT, LF-BK cells were transfected with an LgBiT expression vector, and eight independent LgBiT-transgenic cell lines were obtained (designated #1 to #8). The relative LgBiT protein expression in each LF-BK cell line was detected by Western blotting using a mAb against LgBiT. All cell lines produced LgBiT proteins at similar levels (Figure 3A). To analyze intracellular luminescence signal intensity under stable LgBiT expression, we compared Nano-Luc activity between transiently LgBiT-transfected cells and cells stably expressing LgBiT. As shown in Figure 4B, transiently transfected cells and cells stably expressing LgBiT were infected with wild-type or 10-fold serially diluted inactivated AS-HiBiT and then assessed for Nano-Luc activity. The luminescence intensity gradually decreased in 10-fold serially diluted AS-HiBiT samples and peaked in the undiluted inactivated AS-HiBiT sample (1×). Further, the luciferase activity of stable cells inoculated with 1× inactivated AS-HiBiT was higher than that of transiently transfected cells (Figure 3B). These results suggested that cells stably expressing LgBiT produce strong intracellular luminescence signals.

### 3.4. Inactivated FMDV AS-HiBiT Entry Using Anti-Serum

Guinea pig serum samples collected on 0 and 28 dpv were evaluated for an immune response to FMDV Asia1 using SP ELISA, VNT, and the Nano-Luc assay (Figure 4A–C). The serum samples collected at 0 dpv tested negative for SP Asia1 Ab, with PI values of <20% and VN titers of ≤0.9 log10. The anti-sera samples collected at 28 dpv tested positive for SP Asia1 Ab, with PI values of ≥80% and VN titers of ≥2.558 log10 (Figure 4A,B). Negative and positive sera serologically diagnosed for FMDV Asia1 SP Ab were serologically evaluated using the NanoBiT split luciferase assay (Figure 4C). After entering inactivated FMDV AS-HiBiT (1 × 10^4^ relative luminescence units RLU of Luc) into stable LgBiT-LF-BK cells (2 × 10^4^ cells/well), negative and positive anti-sera diluted 10 to 2000 times were measured using the Nano-Glo Luc assay system. Diluted positive sera could block the entry of inactivated FMDV AS-HiBiT, indicating the presence of neutralizing Abs against FMDV Asia1. The RLU values of positive anti-sera diluted 10 or 100 times were significantly lower than those of negative sera. In contrast, the RLU values of positive sera diluted more than 1000 times did not significantly differ from those of negative sera. Further, there were no significant differences among negative sera diluted 10 to 2000 times. Taken together, these findings indicated that inactivated AS-HiBiT may be useful for the rapid detection of neutralizing Abs in anti-sera against FMDV Asia1 SP.

## 4. Discussion

We demonstrated the effective cellular entry of inactivated FMDV AS-HiBiT using intracellular luminescence measurements and protein analysis. The assay was performed using living cells, without chemical fixation, allowing cellular entry to be monitored. The inactivated FMDV AS-HiBiT entry assay may lead to the development of a new neutralizing Ab assay for the diagnosis and evaluation of FMD. This novel assay has advantages over traditional diagnostic tests, in which the endpoint titer is determined as the reciprocal of the last serum dilution that fully prevents the cytopathic effect (CPE) by 100 TCID50 of FMDV [26]. Traditional VNTs are based on the CPE by virus infection and are conducted in a BSL-3 facility, and results may differ among workers or laboratories as they are visually interpreted. A NanoBiT assay based on FMDV AS-HiBiT entry may produce more consistent VNT results than existing methods, and in a shorter time. 

Highly quantitative detection systems based on split Nano-Luc complementation targeting various viruses have been recently developed. For example, a split Nano-Luc complement-based human norovirus VLP-HiBiT entry assay using 293T-FUT2 (human embryonic kidney) cells stably expressing LgBiT and a split Nano-Luc-based substitutive infection and neutralization assay for the evaluation of candidate human norovirus vaccines at the cellular level have been developed [27,28]. In addition, a split Nano-Luc-based serological assay that detects Abs against SARS-CoV-2 in patient samples has been developed [29]. In this assay, a recombinant receptor-binding domain (RBD) of the SARS-CoV-2 spike glycoprotein fused with HiBiT is incubated with patient blood samples and protein G beads. If RBD-binding Abs are present, RBD-HiBiT/Ab/protein G complexes precipitate and can be detected using LgBiT as a detection reagent [29]. This entry assay facilitates the screening of novel antiviral drug candidates that target the early steps of the FMDV life cycle. An infectious recombinant hepatitis E virus (HEV) harboring the nanoKAZ gene, which has an amino acid sequence identical to that of NanoLuc but a different nucleotide sequence in the hypervariable region of open reading frame 1, termed eHEV-nanoKAZ, is applicable to the screening of antiviral drugs that the target early steps of the HEV life cycle [30]. As for animal viruses, a recombinant porcine reproductive and respiratory syndrome virus (PRRSV) harboring HiBiT inserted in the Nsp2 region (RvJX-Nsp2325-HiBiT) has been developed [31]. The authors demonstrated that RvJX-Nsp2325-HiBiT can be used as a highly efficient platform for the screening and evaluation of anti-PRRSV therapies by showing that RvJX-Nsp2325-HiBiT is a convenient and stable tool for evaluating viral propagation both in vitro and in vivo. In addition, a recombinant senecavirus A (rSVA) expressing the Nano-Luc gene between SVA 2A and 2B has been established [32]. Based on a comparison of rSVA Nano-Luc with the traditional PRNT50 assay, a new neutralization assay was proposed, and the applicability of rSVA Nano-Luc to antiviral interferon-stimulated gene screening was demonstrated. Like FMDV, SVA belongs to the family Picornaviridae; therefore, SVA needs to be differentiated from FMDV. This implies that it is possible to develop a novel neutralization assay using recombinant FMDV. 

The NanoBiT split Luc-based AS-HiBiT entry assay can be used for the comprehensive evaluation of FMDV antiviral drug candidates and seroconversion screening and to develop novel tools for testing FMDV-neutralizing Abs. Additionally, stable LgBiT-LF-BK cell lines and inactivated Asia1 Shamir Abs can be sustainably supplied to laboratories or companies without BSL-3 facilities. Although we did not conduct animal inoculation tests with inactivated AS-HiBiT, if it is proven to be effective as an FMD vaccine, inactivated AS-HiBiT could be utilized as a DIVA (“Differentiating Infected from Vaccinated Animals”) vaccine. In this study, the reaction between inactivated AS-HiBiT and anti-sera was demonstrated, but optimal conditions for VNTs using inactivated AS-HiBiT were not established. If a detailed assay protocol is established for evaluating and measuring neutralizing Abs against inactivated AS-HiBiT, a new surrogate VNT that does not require a BSL-3 facility can be developed. Moreover, if the entry assay is applied to HiBiT-tagged FMDV-like particle (VLP), HiBiT-tagged FMDV VLP could represent a safe and useful tool for detecting neutralizing Abs in serum samples under BSL-2 conditions. In a further study, an HiBiT-tagged Asia1 shamir virus-like particle (AS-HiBiT VLP) has been developed and studies are being conducted for the application of the AS-HiBiT VLP to a novel VNT that can be conducted in BSL-2 facilities. 

In conclusion, we developed a NanoBiT split Luc-based AS-HiBiT entry assay based on viral endocytosis in living cells. This system will facilitate reliable and comprehensive evaluation of FMD vaccines and Ab candidates and provide a useful tool for investigating FMDV cellular entry.

## Figures and Tables

**Figure 1 viruses-16-01002-f001:**
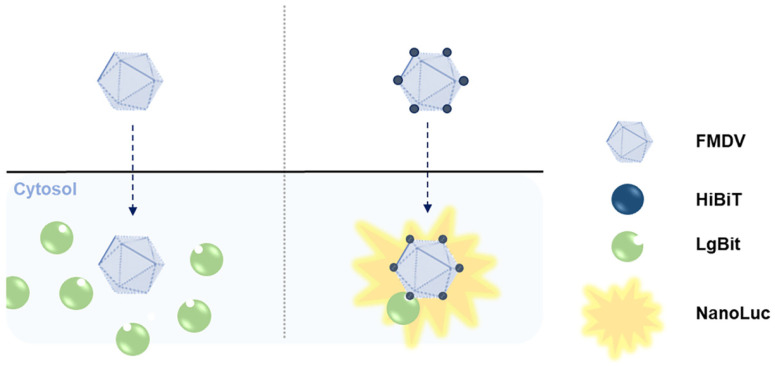
Detection of FMDV entry by the NanoBiT split luciferase assay. Schematic representation of the HiBiT-tagged FMDV and LgBiT-expressing cells.

**Figure 2 viruses-16-01002-f002:**
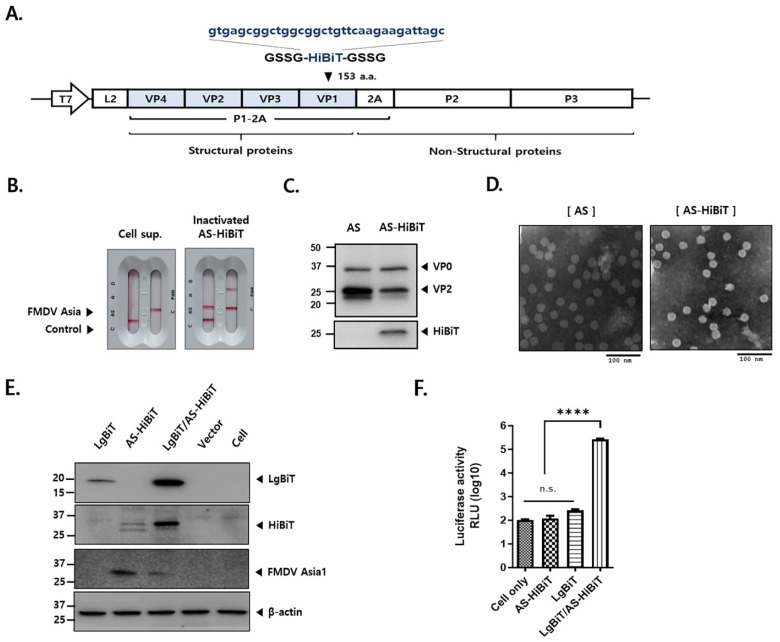
Production and characterization of FMDV Asia1 Shamir-HiBiT (AS-HiBiT). (**A**) Schematic representation of the genome of the FMDV Asia1 Shamir (AS) full-length infectious clone with the HiBiT gene inserted into the VP1 region (AS-HiBiT). (**B**) AS-HiBiT was produced and detected using a VDRG FMDV 3Diff/PAN Ag Rapid kit. (**C**) Western blotting analysis of inactivated FMDV AS and inactivated AS-HiBiT antigen. (**D**) Electron microscopic images of inactivated FMDV AS and inactivated AS-HiBiT. Bar = 100 nm. (**E**) LF-BK cells were transiently transfected with an empty vector or LgBiT expression vector, infected with inactivated AS-HiBiT, and analyzed by Western blotting using appropriate Abs. (**F**) After transfection with empty vector or recombinant plasmid followed by infection with inactivated AS-HiBiT, NanoBiT split luciferase activity was measured using the luciferase assay system. Data are mean ± SEM (*n* = 8/group). **** *p* < 0.0001 and ns, *p* > 0.05 (one-way ANOVA followed by Tukey tests).

**Figure 3 viruses-16-01002-f003:**
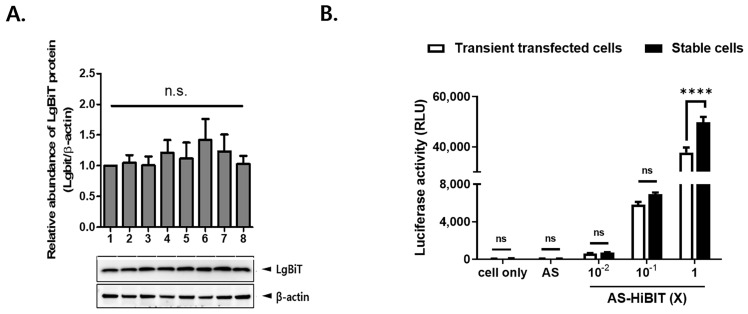
Establishment and characterization of LF-BK cells stably overexpressing LgBiT (LgBiT-LF-BK). (**A**) LF-BK cells were transfected with an LgBiT expression vector, followed by selection with hygromycin B. Eight stable LgBiT-LF-BK cell lines (#1 to #8) were obtained. LgBiT expression levels were determined by Western blotting. LgBiT protein abundances were calculated from band intensities (normalized to β-actin) measured using the Image J software. (**B**) NanoBiT luciferase activity of inactivated AS-HiBiT cell entry was measured in transiently LgBiT-transfected LF-BK cells or stable LgBiT-LF-BK cells. Data are mean ± SEM. **** *p* < 0.0001; ns, *p* > 0.05 (two-way ANOVA followed by Sidak multiple comparison tests).

**Figure 4 viruses-16-01002-f004:**
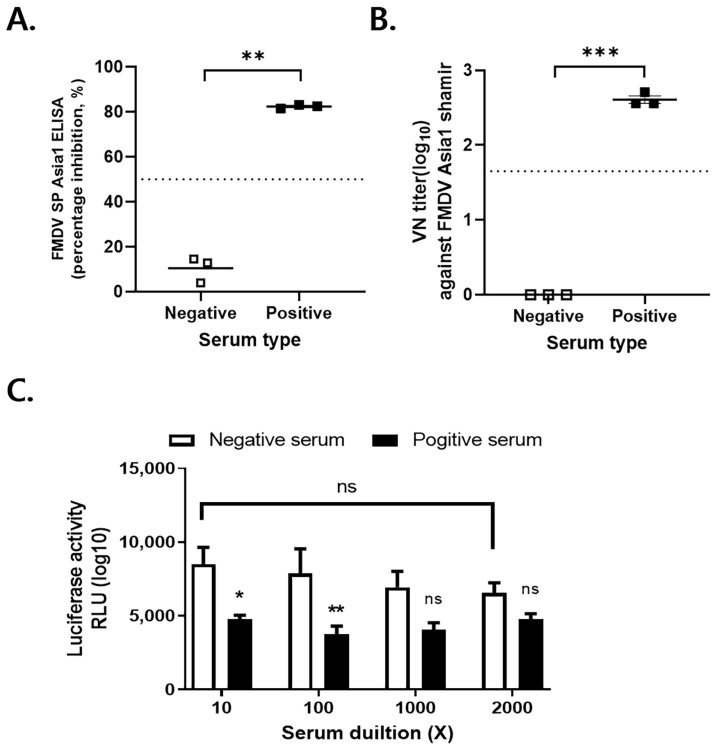
Application of the sensitive Nano-Luc-based biosensor to serological testing of FMDV Asia1. Anti-sera of guinea pigs collected at 0 (negative serum) and 28 (positive serum) dpv were analyzed using FMDV SP Asia1 Ab ELISA, VNT, and the Nano-Luc assay. (**A**) Type Asia1 SP-specific Ab titers by FMDV type Asia1 SP ELISA. A PI value of ≥50% was considered positive for an immune response to FMDV Asia 1. (**B**) VN titers (log10) of ≥1.65 and ≤0.9 were, respectively, considered positive and negative for FMDV Asia 1 Shamir. Data are mean ± SEM (*n* = 3/group). *** *p* < 0.001 and ** *p* < 0.01 (paired *t*-test). (**C**) Inactivated AS-HiBiT was applied to serological testing of FMDV Asia1 Shamir-specific Ab. LgBiT-LF-BK cells were incubated with 10-fold serial diluted serum/AS-HiBiT and then tested for luciferase activity using the NanoBiT assay system. Data are mean ± SEM (*n* = 3/group). * *p* < 0.05; ** *p* < 0.01; and ns, *p* > 0.05 (two-way ANOVA followed by Sidak multiple comparison tests).

## Data Availability

All data generated or analyzed during this study are included in this published article. The datasets were available from the corresponding author on reasonable request.

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
