# Peer review of "Establishment of the Foot-and-Mouth Disease Virus Type Asia1 Expressing the HiBiT Protein: A Useful Tool for a NanoBiT Split Luciferase Assay"

_viruses, 2024, doi:10.3390/v16071002_

Round 1

Reviewer 1 Report

Comments and Suggestions for Authors

Overall I think that this is a nice manuscript for which the authors should be congratulated. Whilst a NanoLuc expressing FMDV has previously been reported, this new version represents an interesting and potentially useful additional reagent. Overall I think that the manuscript is clear and nicely presented. I just have a few minor comments.

Introduction:

Why did the authors select Asia1 when serotype O is the most predominant serotype globally?

Methods:

What is the commercial source of the vaccine used to generate the sera (line 91)?

The authors should provide more details with regards to the infectious clone and its origin. Is there a reference that can be provided?

How well does the recombinant virus grow relative to the wild type?

The HiBiT tag has been added to a significant neutralising antigenic site of the virus, potentially altering the neutralisation characteristics of the virus. Have the authors performed any experiments comparing the neutralisation characteristics of the HiBiT modified vs wild type virus?

Similar to above, have the authors undertaken classical neutralisation assays using the live virus (in IBRS2 cells) to compare the neutralisation titres obtained using their assay relative to the classical assay. This would be important in generating a compelling case for its use more widely and should be reported.

I think that the authors should add the reference to a FMDV expressing nanoluciferase virus (Zhang et al, PMID: 28532601).

Comments on the Quality of English Language

There is an overuse of inaccurate hyphens/dashes. This is minor and should be corrected during editing.

Author Response

Response to Reviewer Comments

First of all, we sincerely thank you for your careful comments to improve the quality of our paper.

Introduction:

  1. Why did the authors select Asia1 when serotype O is the most predominant serotype globally?

- Thank you for your revision. Type Asia1 has not yet occurred in South Korea but occurred intermittently in neighboring countries including China, Mongolia and North Korea (Pool1). Therefore, the risk of type Asia1 introduction into South Korea cannot be excluded. So we chose Asia1 first. For Asia1, the reason for choosing the Shamir strain is because the Shamir vaccine is the only international vaccine lineage in Asia1 and is the Asia1 vaccine recommended by the World Reference Institute for Foot and Mouth Disease (WRLFMD, www.wrlfmd.org). Although not mentioned in this paper, we are currently conducting the similar research as Asia1 Shamir on FMDV types O and A.

Methods:

  1. What is the commercial source of the vaccine used to generate the sera (line 91)?

  • Thank you for your revision. In this study we used Boehringer Ingelheim’s trivalent FMDV vaccine. To ensure clear information, we have been revised the text of the paper as below

“Line 91-92”:

A commercial FMD vaccine (0.2mL/dose) containing O Manisa, O 3039, A22 Iraq, and Asia1 Shamir (Boehringer Ingelheim, RP, DEU).

  1. The authors should provide more details with regards to the infectious clone and its origin. Is there a reference that can be provided?

  • Thank you for your revision. To ensure clear information, we have been revised the text of the paper as below.

“Line 105-107”:

The GSSG linker and HiBiT (ggcagcagcggcVSGWRLFKKISggcagcagcggc) coding sequence were inserted into the VP1 region of Asia1 shamir (GenBank accession No. JF739177) and the modified complete genome of the Asia1 shamir was cloned into the pBluescript SK II vector to generate recombinant FMDV infectious clones (Shin et al, PMID: 32827925).  

  1. How well does the recombinant virus grow relative to the wild type?

  • Thank you for your revision. When titration was performed in LF-BK cells with recombinant virus and live virus and compared, the results were 4.1×106TCID50/ml and 1.1×107TCID50/ml, respectively.

When recombinant virus was inoculated into adherent LF-BK cells ( in 175T flasks), 100% CPE was seen within 20 hours, which is similar to the results observed with live virus. It is believed that recombinant virus also grows well like live virus.     

  1. The HiBiT tag has been added to a significant neutralising antigenic site of the virus, potentially altering the neutralisation characteristics of the virus. Have the authors performed any experiments comparing the neutralisation characteristics of the HiBiT modified vs wild type virus?

  • Thank you for your revision. Although we did not conduct an experiment to compare VN titer after performing VNT with recombinant virus and live virus, as mentioned in the discussion of this paper (lines 335-337), we developed the HiBiT-tagged Asia1 Shamir VLP (with HiBiT inserted at the same position as the HiBiT-tagged Asia1 Shamir) and performed VNT using this VLP and wild-type virus. When their correlation was examined, a high R2 value was shown. Accoriding to this result, the neutralization characteristics of the recombinant virus with HiBiT inserted at the same position are expected to be intact.

  1. Similar to above, have the authors undertaken classical neutralisation assays using the live virus (in IBRS2 cells) to compare the neutralisation titres obtained using their assay relative to the classical assay. This would be important in generating a compelling case for its use more widely and should be reported.

- As the reviewer said, neutralization assays were previously performed in our laboratory using IBRS2 cells. Although the results of VNT using IBRS2 cells and VNT using LF-BK cells were similar, IBRS cells had the tendency of lower virus infectivity when overgrown and recently, there have been reports that LF-BK was originally thought to be of bovine fetal kidney origin but originates from pig [Michael LaRocco et al, 25617444] [Oliver Lung et al, 34737324]. Therefore, we performed VNT using LF-BK cells, but in the next study, we will also perform an experiment to compare titers by performing VNT using IBRS2 cells. Thank you for your comments.

  1. I think that the authors should add the reference to a FMDV expressing nanoluciferase virus (Zhang et al, PMID: 28532601).

  • Thank you for making good point. We have revised the text as below to reflect your comment.

“Line 53-55”:

Nano-luciferase (Nano-Luc)-based biosensors have been applied to several viruses, including the SARS-CoV-2 virus, dengue virus, Japanese encephalitis virus, hepatitis C virus and FMDV as well. [15, Zhang et al, PMID: 28532601]

Reviewer 2 Report

Comments and Suggestions for Authors

The authors of the manuscript entitled: “Establishment of Foot-and-Mouth Disease Virus Type Asia1 Expressing the HiBiT Protein: A Useful Tool for a NanoBiT Split Luciferase Assay by Giyoun Cho etal describes the proof-of-principle assay that can be used for testing of potential FMD antiviral candidates and seroconversion screening outside of a rigorous BSL-3 facility. 

Major points of consideration:

1. The authors should improve the abstract. For example, the following two sentences: “Inactivated AS-HiBiT was entered into LgBiT-LF-BK cells, leading to make luminescence signals. Luminescence signals intensified as cell and virus concentrations increased” should provide more details.  

2. Because the assay is intended to be used to test antiviral candidates and for seroconversion screening, it would be useful for the authors to test the stability of the system at different experimental conditions, such as temperature or pH.  

Minor grammar/structure issues:

Line 17:  The word “limited” is repeated twice in the same sentence.  You could say “experiments on FMDV call for more attention”. “

Line 21: The statement “HiBiT-tagged Asia1 Shamir FMDV (AS-HiBiT)” is not complete.  Provide more description.

Line 54:  It should be including, not in-cluding.

Line 115.  It should be Sigma-Aldrich, not Si-ma-Aldrich.

Line 138:  Please, correct the word “pri-mary”.

Line 282:  Please, correct the word “in-tracellular”.

Comments on the Quality of English Language

Please, fix minor spelling issues, and re-write suggested sentences for clarity.

Author Response

Response to Reviewer Comments

First of all, we sincerely thank you for your careful comments to improve the quality of our paper.

 Major points of consideration:

  1. The authors should improve the abstract. For example, the following two sentences: “Inactivated AS-HiBiT was entered into LgBiT-LF-BK cells, leading to make luminescence signals. Luminescence signals intensified as cell and virus concentrations increased” should provide more details.  
  • Thank you for your revision. To ensure clear information, we have been revised the text of the paper as below.

“Line 53-55”:

It was confirmed that Inactivated AS-HiBiT infected LgBiT-LF-BK cells and produced a luminescence signal by binding to intracellular LgBiT of LgBiT-LF-BK cells. In addition, the luminescence signal became stronger as the number of LgBiT-LF-BK cells increased or the concentration of inactivated AS-HiBiT increased.

  1. Because the assay is intended to be used to test antiviral candidates and for seroconversion screening, it would be useful for the authors to test the stability of the system at different experimental conditions, such as temperature or pH.  

- Thank you for making good point. When conducting further research related to antiviral candidate and seroconversion screening with the recombinant Asia1 shamir we developed, taking your comment into consideration, we will conduct an antigen stability experiment according to temperature changes and pH changes. 

Minor grammar/structure issues:

Line 17:  The word “limited” is repeated twice in the same sentence.  You could say “experiments on FMDV call for more attention”. “

  • Thank you for your revision. we have been revised the text of the paper as below.

“Line 17”:

Given that this high-risk pathogen has to be handled in a biosafety level (BSL)-3 facility for safety reasons and the limited availability of BSL-3 laboratories, experiments on FMDV call for more attention.

Line 21: The statement “HiBiT-tagged Asia1 Shamir FMDV (AS-HiBiT)” is not complete.  Provide more description.

  • Thank you for your revision. we have been revised the text of the paper as below.

“Line 21”:

We developed HiBiT-tagged Asia1 Shamir FMDV (AS-HiBiT), a recombinant Asia1 shamir FMDV with HiBiT attached to the VP1 region of Asia1 shamir FMDV.

Line 54:  It should be including, not in-cluding.

  • Thank you for your revision. we have been revised the text of the paper as below.

“Line 54”:

Nano-luciferase (Nano-Luc)-based biosensors have been applied to several viruses, including the SARS-CoV-2, dengue virus, Japanese encephalitis virus, and hepatitis C virus, but not to FMDV.

Line 115.  It should be Sigma-Aldrich, not Si-ma-Aldrich.

  • Thank you for your revision. we have been revised the text of the paper as below.

“Line 115”:

To inactivated the viruses, 3 mM binary-ethylenimine (Sigma-Aldrich) was added, 

Line 138:  Please, correct the word “pri-mary”.

  • Thank you for your revision. we have been revised the text of the paper as below.

“Line 138”:

Incubated with appropriate primary Abs at 4°Ð¡ under shaking overnight.

Line 282:  Please, correct the word “in-tracellular”.

  • Thank you for your revision. we have been revised the text of the paper as below.

“Line 282”:

We demonstrated effective cellular entry of inactivated FMDV AS-HiBiT using intracellular luminescence measurement and protein analysis.
